# Dickkopf-1 Promotes Angiogenesis and is a Biomarker for Hepatic Stem Cell-like Hepatocellular Carcinoma

**DOI:** 10.3390/ijms23052801

**Published:** 2022-03-03

**Authors:** Tsuyoshi Suda, Taro Yamashita, Hajime Sunagozaka, Hikari Okada, Kouki Nio, Yoshio Sakai, Tatsuya Yamashita, Eishiro Mizukoshi, Masao Honda, Shuichi Kaneko

**Affiliations:** Department of Gastroenterology, Graduate School of Medical Science, Kanazawa University, Kanazawa, Ishikawa 920-8641, Japan; t.suda1112@gmail.com (T.S.); sunagozakah@gmail.com (H.S.); okada0922@gmail.com (H.O.); nio@m-kanazawa.jp (K.N.); yoshios@m-kanazawa.jp (Y.S.); ytatsuya@m-kanazawa.jp (T.Y.); eishirom@m-kanazawa.jp (E.M.); mhonda@m-kanazawa.jp (M.H.); skaneko@m-kanazawa.jp (S.K.)

**Keywords:** hepatocellular carcinoma, Dickkopf-1, angiogenesis, cancer stem cell

## Abstract

Cancer stemness evinces interest owing to the resulting malignancy and poor prognosis. We previously demonstrated that hepatic stem cell-like hepatocellular carcinoma (HpSC-HCC) is associated with high vascular invasion and poor prognosis. Dickkopf-1 (DKK-1), a Wnt signaling regulator, is highly expressed in HpSC-HCC. Here, we assessed the diagnostic and prognostic potential of serum DKK-1. Its levels were significantly higher in 391 patients with HCC compared with 205 patients with chronic liver disease. Receiver operating characteristic curve analysis revealed the optimal cutoff value of DKK-1 to diagnose HCC and predict the 3-year survival as 262.2 and 365.9 pg/mL, respectively. HCC patients with high-serum DKK-1 levels showed poor prognosis. We evaluated the effects of anti-DKK-1 antibody treatment on tumor growth in vivo and of recombinant DKK-1 on cell proliferation, invasion, and angiogenesis in vitro. DKK-1 knockdown decreased cancer cell proliferation, migration, and invasion. DKK-1 supplementation promoted angiogenesis in vitro; this effect was abolished by an anti-DKK-1 antibody. Co-injection of the anti-DKK-1 antibody with Huh7 cells inhibited their growth in NOD/SCID mice. Thus, DKK-1 promotes proliferation, migration, and invasion of HCC cells and activates angiogenesis in vascular endothelial cells. DKK-1 is a prognostic biomarker for HCC and a functional molecule for targeted therapy.

## 1. Introduction

Hepatocellular carcinoma (HCC) is the most common type of primary liver cancer and occurs in people with chronic liver disease (CLD). It is the fourth leading cause of cancer-related deaths worldwide [1]. GLOBOCAN data revealed that approximately 841,000 new cases of liver cancer and 782,000 related deaths were reported in 2018, which makes liver cancer the sixth most commonly diagnosed cancer [2]. The global incidence of HCC is heterogeneous due to the varying prevalence of the underlying risk factors. It has been estimated that 72% of the cases occur in Asia (>50% in China), 10% in Europe, 7.8% in Africa, 5.1% in North America, 4.6% in Latin America, and 0.5% in Oceania [3]. Moreover, limited medical and social care resources may play a crucial role in the development of HCC [1]. The incidence of HCC is presumed to increase over the next 10–20 years, with 5-year survival rates ranging from 50% to 75% in the early stages to as low as 3% in patients with distant metastases [4,5].

Hepatitis B and C virus infection, alcohol use, nonalcoholic fatty liver disease, Budd–Chiari syndrome, and aflatoxin are the known risk factors for HCC. In clinical practice, quantitation of serum alpha-fetoprotein (AFP) level and ultrasonography is widely used for the early detection of HCC [6]; however, the sensitivity and specificity of AFP detection at a cutoff value of 20 ng/mL are 59% and 90%, respectively [7]. Thus, novel biomarkers are required to facilitate a better diagnosis of HCC [8]. Biomarkers for treatment are also currently being explored [9,10].

Dickkopf-1 (DKK-1), a soluble secreted protein with low molecular weight, plays a pivotal role in head induction and head embryogenesis in *Xenopus* [11]. DKK-1 negatively regulates the Wnt/β-catenin pathway and competitively binds to LRP5/6, which shows a higher affinity for DKK-1 than Wnt ligands, thereby inhibiting the formation of the Fz-Wnt-LRP5/6 complex and interfering with Wnt signaling [11,12,13]. DKK-1 is associated with carcinogenesis, metastasis, recurrence, and poor prognosis in HCC [14,15,16,17].

Using comprehensive genetic analysis, we previously classified HCC into two types: stem cell type (hepatic stem cell/hepatoblast-HCC, HpSC-HCC), characterized by epithelial cell adhesion molecule (EpCAM) and AFP positivity, and hepatocyte type (mature hepatocyte-HCC, MH-HCC), characterized by differentiated hepatocyte marker positivity [18]. We previously demonstrated that hepatic stem cell markers, EpCAM, AFP, DKK-1, DLK1, CD133, and CK19, were upregulated in HpSC-HCC compared with that in MH-HCC. Moreover, HpSC-HCC is associated with a high frequency of vascular invasion and poor prognosis [19]. We also found that DKK-1 is highly expressed in HpSC-HCC and presented it as a novel biomarker that regulates Wnt signaling [18,19]. DKK-1 expression has also been reported to be elevated in HCC tissues [20], and DKK-1 is dysregulated in various other malignant tumors, such as pancreatic cancer, colorectal cancer, multiple myeloma, and chronic lymphocytic leukemia [21,22,23,24]. The diagnostic value of serum DKK-1 levels for HCC has previously been determined [25,26,27,28,29,30,31,32,33]. However, although DKK-1 expression is activated in HpSC-HCC with the activation of Wnt signaling, DKK-1 itself is known to inhibit the Wnt signaling pathway, and the role of DKK-1 expression in the process of HpSC-HCC development remains elusive.

In the present study, we assessed the diagnostic and prognostic value of serum DKK-1 levels in HCC using clinical samples. We further investigated the effects of DKK-1 on cancer cell proliferation, invasion, and angiogenesis in vitro and of anti-DKK-1 antibody treatment on tumor growth in vivo.

## 2. Results

### 2.1. Elevation of Serum DKK-1 Levels in HpSC-HCC

We re-analyzed the hierarchical clustering of 156 HCC samples using a microarray data set of HpSC-HCC (*n* = 60) and MH-HCC (*n* = 96). We found that 793 genes were significantly differentially expressed between HpSC-HCC and MH-HCC (*p* < 0.001) (Figure 1A). Of these, the genes encoding EpCAM, AFP, and DKK-1 in HpSC-HCC were highly expressed. As EpCAM is a marker of liver cancer stem cells, we further evaluated the expression of genes encoding EpCAM and DKK-1 using EpCAM^+/−^ cells sorted from Huh7 cells. The expression of *DKK-1* was higher in isolated EpCAM^+^ cancer stem cells compared with that in EpCAM^−^ cells (Figure 1B). This pattern of DKK-1 expression was confirmed using Western blot analysis (Figure 1C). We further evaluated the expression of *DKK-1* using surgically resected HCC specimens. Among 58 HCC specimens (14 HpSC-HCC and 44 MH-HCC), *DKK-1* expression could be evaluated in 8 HpSC-HCC and 36 MH-HCC cases by qRT-PCR. We found that *DKK-1* expression was upregulated in HpSC-HCC compared with that in MH-HCC (*p* = 0.002, Figure 1D). Serum DKK-1 levels could be further evaluated in 11 HpSC-HCC and 36 MH-HCC cases by enzyme-linked immunosorbent assay (ELISA) and were found to be higher in HpSC-HCC samples compared to MH-HCC samples (*p* = 0.003, Figure 1E). HpSC-HCC significantly correlated with poor overall and recurrence-free survival compared with MH-HCC (*p* = 0.0005, Figure 1F and *p* = 0.008, Figure 1G). Thus, using the microarray data set and a small HCC cohort, we found that *DKK-1* expression and serum DKK-1 levels were elevated in HpSC-HCC with poor survival outcomes.

### 2.2. Serum DKK-1 as a Diagnostic Marker for HCC

Next, we evaluated the diagnostic and prognostic utility of serum DKK-1 levels by investigating in a relatively large cohort. The clinical characteristics of the patients with HCC enrolled in this study are summarized in Table 1 (*n* = 391). HCV infection is the most common cause of HCC, accounting for approximately 55.5% of HCC cases. Liver function tests showed that 73.1% and 62.7% of the patients were classified as Child–Pugh A and albumin-bilirubin (ALBI) grade 1/2a, respectively. A total of 75% of the patients were diagnosed with Union for International Cancer Control (UICC) stage I/II, and 52% were diagnosed with Barcelona Clinic Liver Cancer (BCLC) stage 0/A cancer. The median serum DKK-1 level was 330.8 pg/mL. The median serum AFP and protein induced by vitamin K absence-II (PIVKA-II) values were 24.0 ng/mL and 50.0 mAU/mL, respectively.

The characteristics of patients with CLD are shown in Table 2 (*n* = 205). Serum DKK-1 levels were significantly higher in HCC (median, 330.8 pg/mL) than in CLD (median, 253.8 pg/mL) (Figure 2A). We evaluated the receiver operating characteristic (ROC) curve of the serum DKK-1, AFP, and PIVKA-II levels in 391 patients with HCC and in 205 patients with CLD and determined the optimal cutoff values for the diagnosis of HCC to be 262.2 pg/mL, 24.5 ng/mL, and 43.5 mAU/mL, respectively (Figure 2B). The sensitivity of the diagnostic evaluation of DKK-1, AFP, and PIVKA-II levels was 80.5%, 49.0%, and 53.7%, respectively. The specificity of the diagnostic evaluation of DKK-1, AFP, and PIVKA-II levels was 53.2%, 91.1%, and 97.2%, respectively. The area under the ROC curve (AUC) of serum DKK-1, AFP, and PIVKA-II levels was 0.708 (95% confidence interval (CI), 0.664–0.752), 0.750 (95% CI, 0.711–0.789), and 0.756 (95% CI, 0.717–0.795), respectively (Figure 2B).

We analyzed the correlation between serum DKK-1, AFP, and PIVKA-II levels in HCC cases using linear regression curve analysis. The correlation (r) between serum DKK-1 and serum AFP levels was 0.44 (*p* < 0.001; Figure 2C), indicating a weak positive correlation, whereas that between serum DKK-1 and serum PIVKA-II levels was 0.27 (*p* < 0.001; Figure 2D).

### 2.3. Serum DKK-1 Levels and Clinicopathological Characteristics of HCC

The serum DKK-1 level was elevated according to HCC size (*p* < 0.001, Figure 3A). In contrast, no difference was observed between solitary (median, 333.8 pg/mL) and multinodular (median, 327.9 pg/mL) HCC cases (Figure 3B). Serum DKK-1 levels were higher in the massive/diffuse type (median, 446.4 pg/mL) than in the nodular type (median, 323.3 pg/mL) (*p* < 0.001; Figure 3C), according to Eggel’s classification.

Serum DKK-1 levels were elevated in portal vein invasion (Vp)-positive patients (median, 445.9 pg/mL) compared with those in Vp-negative patients (median, 326.8 pg/mL) (*p* < 0.001; Figure 3D). In particular, the serum DKK-1 levels were elevated in Vp3/4 patients compared with those in Vp- or Vp1/2 patients (*p* < 0.001, Vp3/4: median, 487.8 pg/mL; Vp1/2: median, 391.8 pg/mL; Vp-: median, 326.8 pg/mL) (Figure 3E). Serum DKK-1 levels were significantly higher in patients in advanced stages classified by the BCLC and UICC (*p* < 0.001) (Figure 3F, G). Serum DKK-1 levels were not related to the hepatic reserve in terms of the Child–Pugh score and ALBI grade (Figure 3H, I). Taken together, the above data indicated that the serum DKK-1 level was elevated in HCC with a proliferative and invasive nature characterized by advanced stages with a high frequency of Vp.

### 2.4. Serum DKK-1 Level as a Prognostic Marker of HCC

We evaluated the prognostic utility of serum DKK-1 levels in 271 cases with available 3-year survival information. From the ROC curve analysis, a cutoff value of 365.9 pg/mL was determined (AUC 0.601, sensitivity 71.9%, specificity 52.9%; Figure 4A) to differentiate between patients with HCC who survived for more than 3 years and those who did not. Kaplan–Meier survival analysis indicated that elevated serum DKK-1 levels (≥365.9 pg/mL) were significantly correlated with poorer prognosis (*p* < 0.001, hazard ratio (HR) 2.40 (95% CI 1.59–3.62)) (Figure 4B).

The characteristics of patients with/without an elevated serum DKK-1 level (365.9 pg/mL as cutoff) are presented in Table 3. Significant differences were observed in sex, etiology of liver diseases, UICC stage, and BCLC stage between the two groups.

We evaluated the effect of serum DKK-1 levels on prognostic stratification according to the hepatic reserve. In both ALBI grade 1/2a (good hepatic reserve) and 2b/3 (poor hepatic reserve) cohorts, the high-serum DKK-1 group showed a significantly worse prognosis (ALBI grade 1/2a; *p* < 0.001, HR 3.29, 95% CI 1.81–5.97; Figure 4C) (ALBI grade 2b/3. For ALBI grade 2b/3, *p* < 0.001, HR 3.59, 95% CI 1.79–7.20; Figure 4D).

The prognosis was further evaluated in early-stage patients with HCC who received surgery or in those with advanced-stage HCC who received systemic therapy. The prognosis was significantly worse in patients with elevated DKK-1 levels in both early (*p* = 0.045, HR 6.55, 95% CI 1.49–28.8; Figure 4E) and advanced (*p* = 0.024, HR 2.49, 95% CI 1.29–4.79; Figure 4F) stages.

We performed univariate and multivariate analyses using the Cox proportional hazards model (Table 4). The cutoff value of serum DKK-1 level was determined as 365.9 ng/mL, as described above. The cutoff value of serum AFP and PIVKA-II levels was set as 400 mg/mL and 90 mAU/mL, respectively, according to previous studies [34,35,36,37,38]. Univariate analysis indicated that hepatic reserves (Child–Pugh classes and ALBI grades), tumor stages (UICC and BCLC stages), and tumor markers (DKK-1, AFP, and PIVKA-II levels) were risk factors for poor survival outcomes. Multivariate analysis indicated that ALBI grades, UICC stages, DKK-1 levels, and PIVKA-II levels were independent prognostic factors for poor survival outcomes in HCC patients.

We also examined the effect of treatment in patients who were subsequently treated with chemotherapy or molecular targeted therapy using sorafenib on the level of DKK-1 at the time of initial treatment. For this analysis, 94 patients subjected to chemotherapy and 41 patients subjected to molecular targeted therapy were selected. The serum DKK-1 levels were higher in patients with progressive disease than in those with complete response/partial response/stable disease in the chemotherapy (*p* = 0.002; Appendix A) and molecularly targeted therapy (*p* = 0.023; Appendix A) groups.

### 2.5. DKK-1 Expression in HCC Tissues is Correlated with Recurrence

To clarify whether the increase in serum DKK-1 levels could really be attributed to DKK-1 production in HCC cases with poor prognosis, we evaluated the expression of DKK-1 using immunohistochemistry (IHC) in 58 HCC samples obtained from patients who received surgery. DKK-1 expression was classified as low (*n* = 28) or high (*n* = 30) as described in Section 4.4 (Figure 5A,B). The Kaplan–Meier survival analysis indicated that patients with high DKK-1 levels showed significantly worse recurrence-free survival than those with low DKK-1 levels (*p* = 0.026, HR 3.07, 95% CI 1.18–8.00: Figure 5C). The overall survival rate did not differ between the high- and low-DKK-1 groups (Figure 5D), most likely due to the small sample size and short observation period.

### 2.6. The Role of DKK-1 in HCC Growth

The expression of *DKK-1* mRNA was evaluated in EpCAM-positive HuH7 and Hep3B and EpCAM-negative HLE and HLF HCC cell lines. The expression was higher in Huh7 and Hep3B cells than in HLE and HLF cell lines (Figure 6A). Similarly, DKK-1 protein levels in the culture supernatant were significantly higher in Huh7 and Hep3B cells than in HLE and HLF cell lines, as evaluated by ELISA (Figure 6B).

We knocked down *DKK-1* using small interfering RNA (siRNA) in Huh7 and Hep3B cells, and the effect was confirmed at the mRNA and protein levels (*p* < 0.001; Figure 6C, D). The proliferation of Huh7 and Hep3B cells was significantly suppressed by *DKK-1* knockdown (*p* < 0.01; Figure 6E). Similarly, migration and invasion capacity were significantly or almost significantly attenuated by *DKK-1* knockdown in Huh7 and Hep3B cells (*p* = 0.017 and *p* < 0.001, respectively; Figure 6F) (*p* = 0.069 and *p* = 0.0013, respectively; Figure 6G).

We further evaluated the effect of DKK-1 on angiogenesis and performed a tube formation assay using human umbilical vein endothelial cells (HUVECs) and time-lapse imaging. Supplementation with recombinant DKK-1 slightly accelerated tube formation over that of the controls, whereas administration of anti-DKK-1 antibody inhibited tube formation (Figure 7A; time-lapse videos are available in Appendix A). Taken together, these data indicate that DKK-1 not only promotes cell proliferation and invasion in cancer cells but also activates the angiogenesis pathway in vascular endothelial cells. 

### 2.7. Anti-DKK-1 Antibody as a Treatment Option for EpCAM-Positive DKK-1-Positive HCC

Finally, we tested whether DKK-1 could be a molecular target for EpCAM-positive DKK-1-positive HCC with a poor prognosis. We subcutaneously injected Huh7 cells in NOD/SCID mice with anti-DKK-1 neutralizing antibodies or control IgG. Four weeks after transplantation, mice were euthanized, and tumor volumes were evaluated. Anti-DKK-1 treatment clearly suppressed HCC growth compared with control IgG (Figure 7B,C). These data clearly suggest the utility of anti-DKK-1 neutralizing antibodies for the treatment of liver cancer with high DKK-1 expression.

## 3. Discussion

Carcinogenesis and embryogenesis share common features in terms of active cell proliferation, cell motility, stromal cell interaction, and stem cell presence [39,40]. We previously demonstrated that aggressive HCC can be classified according to certain developmental stages of the liver based on the expression status of the stem cell markers, EpCAM, and AFP [18]. DKK-1, a Wnt regulator activated during embryogenesis, was previously shown to be one of the EpCAM-coregulated genes whose expression was activated in HpSC-HCC [19], and in the present study, we identified that serum DKK-1 levels could be used to diagnose HCC with poor survival outcome. We previously demonstrated that AFP elevation is accompanied by a high KI-67 labeling index in surgically resected HCC specimens [41], which might be explained by the current findings of cell proliferation induced by DKK-1. Furthermore, we determined that DKK-1 enhances angiogenesis in vitro, and anti-DKK-1 neutralizing antibody inhibits tumor growth in vivo. Thus, DKK-1 is a biomarker for the diagnosis of aggressive HCC and a potential therapeutic target to inhibit tumor growth.

The utility of serum DKK-1 level for the diagnosis of hepatitis B virus (HBV)-related HCC was previously reported [25]. A meta-analysis revealed that DKK-1 has higher sensitivity and specificity than AFP for HCC diagnosis, and the sensitivity and specificity of serum DKK-1 were 72% and 86%, whereas those of serum AFP were 62% and 82%, respectively [33]. In contrast, we found the sensitivity and specificity of DKK-1 for HCC diagnosis to be 80.5% and 53.2%, at best (cutoff: 262.2 pg/mL). Although it is still unclear why the specificity of serum DKK-1 was lower in our study compared with that in previously published ones, a possible explanation is that DKK-1 may be produced not only in HCC tissues but also in adjacent non-cancerous liver tissues. Indeed, we found that the immunostaining of DKK-1 was positive in adjacent non-cancerous hepatocytes in resected HCC specimens (data not shown), which may result in false positivity in patients with CLD without HCC. Therefore, measuring serum DKK-1 levels may be more relevant for the diagnosis of aggressive HCC with stem cell features and poor prognosis when the DKK-1 levels are highly elevated. Further studies are required to set the appropriate cutoff values with an optimized ELISA system.

Serum DKK-1 levels are elevated in HCC with vascular invasion. Furthermore, DKK-1 promoted tube formation in vitro. The role of DKK-1 in HCC cell proliferation, migration, and invasion has been previously demonstrated [30,42,43,44]. Although DKK-1 is a small protein activated by Wnt signaling, it also inhibits Wnt signaling by binding to the Wnt receptors LRP5/6 [13]. Thus, the role of DKK-1 in Wnt signaling in HpSC-HCC is complex and remains to be elucidated. In the present study, we found that DKK-1 produced by HpSC-HCC potentially activates vascular endothelial cells. Indeed, a previous study revealed the inhibitory effect of Wnt on angiogenesis [45]. It is possible that secreted DKK-1 activates the angiogenesis pathway by inhibiting Wnt signaling in a paracrine manner. Thus, DKK-1 may be a novel therapeutic target to inhibit cancer cell proliferation, migration, invasion, and angiogenesis. Since an anti-DKK-1 neutralizing antibody has already been developed and investigated for the treatment of myeloma [46,47], future studies could evaluate the effect of this antibody in patients with HCC.

The results of this study suggest that serum DKK-1 affects angiogenesis in the microenvironment of HCC. In particular, although AFP is currently used as the first biomarker for identifying HCC patients most likely to receive the survival benefits of ramucirumab therapy [35], the molecular mechanism underlying the correlation of expression status of AFP with the anti-tumor effects of VEGFR2 blockade remains unclear. As *DKK-1* expression correlated positively with that of *AFP* in tumors and is involved in angiogenesis in the tumor microenvironment, serum AFP may be a surrogate marker of DKK-1, which facilitates angiogenesis in HCC. In addition, another angiogenesis inhibitor, bevacizumab, which is used in combination with atezolizumab [36], and the multi-targeted tyrosine kinase inhibitors (TKIs), such as sorafenib, lenvatinib, regorafenib, and cabozentinib [48], also inhibit VEGF and VEGFRs. The relationship between these drugs and DKK-1 needs to be evaluated in the future.

This study has some limitations. The prognostic value of DKK-1 was evaluated in a retrospective manner, and the sample size of the cohorts was relatively small. The serum DKK-1 levels quantified by ELISA could be more specifically and sensitively measured if a chemiluminescent immunoassay system is appropriately developed. DKK-1 and cytoplasmic/nuclear β-catenin status could not be directly evaluated simultaneously due to the limited availability of small FFPE samples. Further studies are required to improve our knowledge regarding DKK-1 biology in HCC.

## 4. Materials and Methods

### 4.1. Patients and Measurement of Serum DKK-1 Levels

Serum DKK-1 levels were measured by ELISA using stored sera from 400 patients with HCC and 205 patients with CLD without HCC as control at the Kanazawa University Hospital from October 2002 to October 2017. Surgically resected HCC samples previously used [41] were evaluated for the expression of EpCAM and AFP as previously described [18]. We included 14 and 44 HCC samples of HpSC-HCC and MH-HCC, respectively, in the study. Among them, RNA samples and sera were available for 8 HpSC-HCC and 35 MH-HCC or 14 HpSC-HCC and 36 MH-HCC specimens, respectively. The Human Dkk-1 Quantikine ELISA Kit (R&D Systems, Minneapolis, MN, USA) was used for the measurement of serum DKK-1 levels. The final value of the DKK-1 level was obtained after correction using a standard curve. The clinical information of the patients was collected retrospectively from medical records. The study conformed to the standards set by the Declaration of Helsinki, and the protocol was approved by the institutional review board of the Graduate School of Medical Sciences, Kanazawa University (IRB number: 2016-093).

### 4.2. Microarray Analysis

A microarray data set of 156 HCC samples (60 HpSC-HCC and 96 MH-HCC) was constructed from a publicly available data set (GEO accession number: GSE5975). Genes differentially expressed between HpSC-HCC and MH-HCC (793 genes) were obtained by a class-comparison analysis with univariate *t*-tests and a global permutation test (1000×) using a BRB-ArrayTools software (version 4.3.2).

### 4.3. Western Blot Analysis

Cells were lysed in radioimmunoprecipitation assay buffer as described previously [49]. The primary antibodies used for Western blot analysis were anti-DKK-1 monoclonal antibody (M11), clone 2A5 (Abnova), and anti–β-actin antibody (Cell Signaling Technology, Inc., Danvers, MA, USA) as per protocol. Immune complexes were visualized using the enhanced chemiluminescence detection reagents (Amersham Biosciences Corp., Piscataway, NJ, USA) as per the manufacturer’s instructions.

### 4.4. Immunohistochemical Staining

Formalin-fixed, paraffin-embedded tissues were prepared for immunohistochemical staining. After deparaffinization, rehydration, antigen retrieval, and blocking (Protein Block Serum-Free; Dako, Carpinteria, CA, USA), the slides were incubated with primary antibodies overnight at 4 °C. The slides were processed using Envision+ Kits (Dako) according to the manufacturer’s instructions. Anti-DKK-1 polyclonal antibody (ab61034, Abcam) was used as the primary antibody. IHC images were analyzed as described previously [18]. Briefly, the staining area was evaluated and scored on four levels (none = 0, focal = 1, multifocal = 2, and diffuse = 3). The staining intensity was evaluated and scored on three levels (none = 0, mild = 1, and strong = 2), and the sum of both levels was used as the immunostaining score (0–5) [50]. The expression of DKK-1 in the tumor was defined as low (≤2 points) or high (≥3 points).

### 4.5. Cell Lines and Reagents

The HCC cell lines, Huh7, Huh1, HLE, HLF, and Hep3B, were supplied by the Japanese Collection of Research Bioresources Cell Bank (Osaka, Japan) or the American Type Culture Collection (Manassas, VA, USA). These cells were maintained at 37 °C in Dulbecco’s modified Eagle’s medium (DMEM; Gibco, Grand Island, NY, USA) supplemented with 10% fetal bovine serum (Gibco).

### 4.6. Cell Sorting

Huh7 cells were trypsinized, washed, and resuspended in Hank’s balanced salt solution (Lonza, Basel, Switzerland) supplemented with 1% HEPES and 2% PBS. Cells were incubated with the fluorescein isothiocyanate-conjugated anti-EpCAM monoclonal antibody BER-EP4 (DAKO) on ice for 30 min prior to cell sorting using FACSAriaII (BD Biosciences, San Jose, CA, USA). Sorted cells were harvested on dishes and cultured overnight for PCR analysis.

### 4.7. Real-Time Quantitative PCR

Total RNA was isolated using the High Pure RNA Isolation Kit (Roche Diagnostics K.K., Tokyo, Japan) according to the manufacturer’s instructions. Quantitative PCR probes for DKK-1 (Hs00183740_m1) were procured from Applied Biosystems (Foster City, CA, USA). The expression of the selected genes was measured in triplicate using a 7900 Sequence Detection System (Applied Biosystems). Each sample was standardized to the normal expression level of the reference β-actin or 18S RNA gene.

### 4.8. RNA Interference

siRNAs specific for human DKK-1 (HSS117947) and a nonspecific control siRNA (scramble) were purchased from Thermo Scientific (Waltham, MA, USA). Cells grown to 60%–80% confluency in 6-well plates were transfected using Lipofectamine^®^ 2000 (Invitrogen Life Technologies, Carlsbad, CA, USA). DKK-1 and scrambled siRNA (60 μmol) were transfected into the cells using Lipofectamine^®^ 2000 according to the manufacturer’s instructions. All experiments were performed 24 h after the transfection. Total RNA was extracted 48 h after transfection, and protein was extracted 72 h after transfection.

### 4.9. Cell Proliferation Assay

In the cell proliferation assay, a single-cell suspension of approximately 1.0 × 10^5^ cells was seeded into a 96-well plate, and the cell density was evaluated at 24, 48, and 72 h after seeding using Cell Counting Kit-8 (Dojindo Research Institute, Kumamoto, Japan), following the manufacturer’s instructions.

### 4.10. Transwell Invasion/Migration Assay

Transwell invasion/migration assay was conducted according to the manufacturer’s protocol using BioCoat Matrigel Invasion Chamber, Cell Culture Inserts, and Control Inserts (Corning). Approximately 1.0 × 10^4^ cells were seeded in the insert chamber and incubated at 37 °C for 48 h. The insert chamber membranes were then fixed with ice-cold methanol and stained with hematoxylin and eosin.

### 4.11. Tube Formation Assay

HUVECs were labeled with the lipophilic fluorescence tracer, dioctadecyloxacarbocyanine perchlorate (DiO). The 8-well chamber slides were coated with Matrigel (BD Biosciences), and approximately 2.5 ×10^3^ HUVECs/well were harvested immediately with Endothelial Cell Growth Basal Medium (Lonza Bioscience) supplemented with recombinant DKK-1 (100 ng/mL), control IgG (15 mg/mL), or anti-DKK-1 neutralizing antibody (15 mg/mL) (R&D Systems). Cells were cultured at 37 °C in 5% CO_2_, and time-lapse images were captured for 48 h using a CSU-X1 spinning disk confocal (Yokogawa, Tokyo, Japan) and Andor iXon3 EMCCD camera system (Andor Technology, Belfast, UK). Images were analyzed using the iQ software (Andor Technology).

### 4.12. Animal Studies

NOD.CB17-Prkdcscid/J (NOD/SCID) male mice were purchased from Charles River Laboratories, Inc. (Wilmington, MA, USA). Mice were housed under specific pathogen-free conditions with a 12 h light/dark cycle and provided ad libitum access to tap water and food. Huh7 cells (approximately 1.0 × 10^6^ cells) were resuspended in 200 μL of a 1:1 DMEM:Matrigel (BD Biosciences) mixture with control IgG (*n* = 3, 100 mg/10^6^ cells) or anti-DKK-1 neutralizing antibody (*n* = 3, 100 mg/10^6^ cells) and subcutaneously injected into 6-week-old NOD/SCID mice. Mice were euthanized, and the tumor volume was evaluated on day 28 after xenotransplantation. The experimental protocol was approved by the Kanazawa University Animal Care and Use Committee and conformed to the Guide for the Care and Use of Laboratory Animals prepared by the National Academy of Sciences.

### 4.13. Statistical Analyses

ROC curves, Mann–Whitney’s *U* test, Kruskal–Wallis test, chi-square test, and log-rank test were performed using GraphPad Prism 7 (GraphPad Software, San Diego, CA, USA). The Cox proportional hazard model was performed using EZR (Saitama Medical Center, Jichi Medical University, Saitama, Japan), a graphical user interface for R (The R Foundation for Statistical Computing, Vienna, Austria). Significance was set at *p* < 0.05. Significance is indicated by *, **, and *** for *p*-values <0.05, <0.01, and <0.001, respectively.

## Figures and Tables

**Figure 1 ijms-23-02801-f001:**
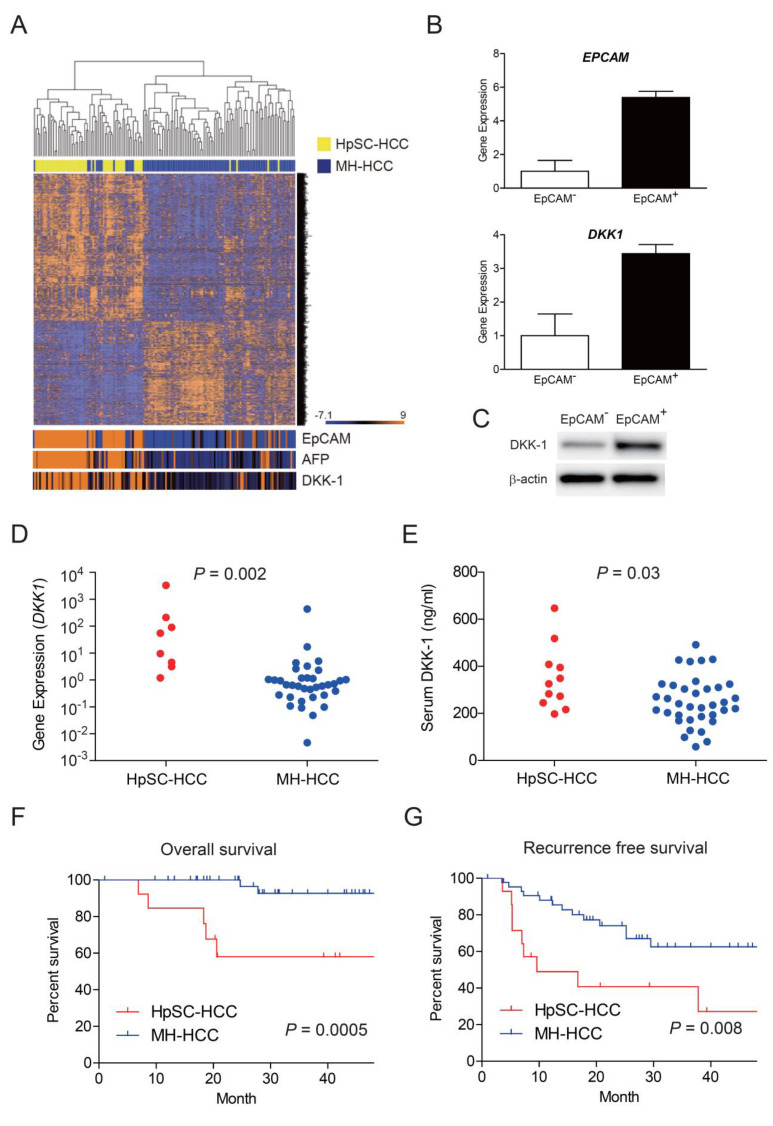
Activation of Dickkopf-1 (DKK-1) in hepatic stem cell-like hepatocellular carcinoma (HpSC-HCC). (**A**) Heatmap images of 793 genes differentially expressed between HpSC-HCC and mature hepatocyte-HCC (MH-HCC). HpSC-HCC and MH-HCC cases are indicated as yellow or blue boxes, respectively. Each cell in the matrix represents the expression level of a gene in an individual sample. Orange and blue cells depict high and low expression levels, respectively, as indicated by the scale bar. Genes encoding epithelial cell adhesion molecule (EpCAM), alpha-fetoprotein (AFP), and DKK-1 were activated in HpSC-HCC. (**B**) qRT-PCR analysis of genes encoding EpCAM and DKK-1 in sorted EpCAM^+/−^ Huh7 cells. (**C**) DKK-1 levels were higher in EpCAM^+^ cells than in EpCAM^−^ cells as assessed by Western blot analysis (**D**) qRT-PCR analysis of *DKK-1* expression in HpSC-HCC (red circle) and MH-HCC (blue circle). (**E**) Serum DKK-1 levels in HpSC-HCC (red circle) and MH-HCC (blue circle). (**F**) Kaplan–Meier curves of overall survival in HpSC-HCC (red line) and MH-HCC (blue line). (**G**) Kaplan–Meier curves of recurrence-free survival in HpSC-HCC (red line) and MH-HCC (blue line).

**Figure 2 ijms-23-02801-f002:**
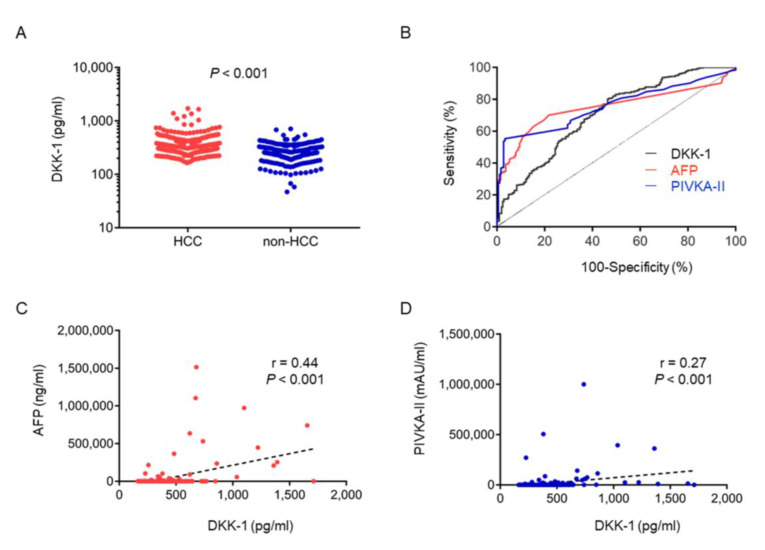
Serum Dickkopf-1 (DKK-1) levels as a diagnostic marker for hepatocellular carcinoma (HCC). (**A**) Serum DKK-1 levels were significantly higher in patients with HCC than in those without HCC. (**B**) A receiver operating characteristic (ROC) curve of serum DKK-1, alpha-fetoprotein (AFP), and protein induced by vitamin K absence-II (PIVKA-II) levels. The optimal cutoff value of serum DKK-1 levels for the diagnosis of HCC was 262.2 pg/mL. The area under the ROC curve of serum DKK-1 levels was 0.708. The sensitivity and specificity of DKK-1 were 80.5% and 53.2%, respectively. (**C**,**D**) The correlation (r) between serum DKK-1 and serum AFP levels was 0.44 (*p* < 0.001; **C**), whereas that between serum DKK-1 and serum PIVKA-II levels was 0.27 (*p* < 0.001; **D**).

**Figure 3 ijms-23-02801-f003:**
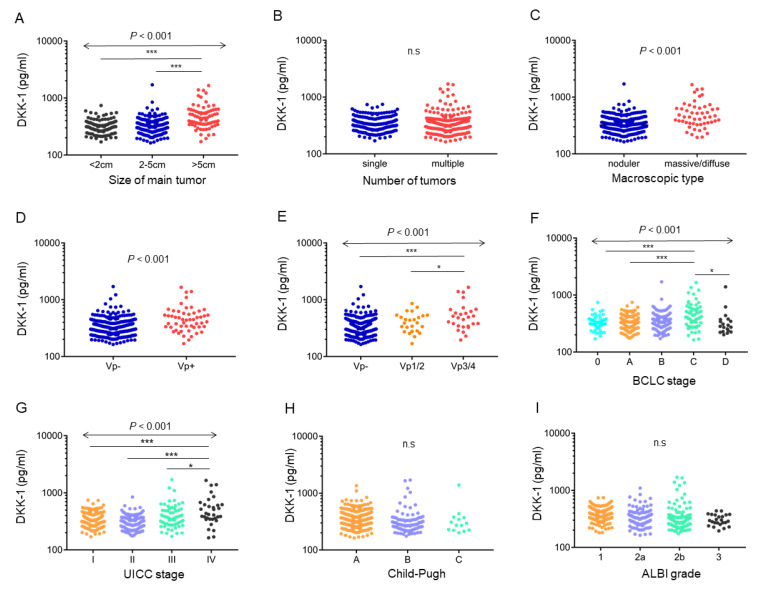
Relationship between serum Dickkopf-1 (DKK-1) levels and clinicopathological characteristics of hepatocellular carcinoma (HCC) and liver function. (**A**) Serum DKK-1 levels were elevated according to HCC size (*p* < 0.001). (**B**) No difference was observed between solitary and multinodular HCC cases. (**C**) Serum DKK-1 levels were higher in the massive/diffuse type than in the nodular type, according to Eggel’s classification. (**D**,**E**) Serum DKK-1 levels were elevated in portal vein invasion (Vp)-positive patients compared with that in Vp-negative patients (*p* < 0.001; D). Serum DKK-1 levels were elevated in Vp3/4 patients compared with those in Vp- or Vp1/2 patients (*** *p* < 0.001 and * *p* < 0.05; **E**). (**F**,**G**) Serum DKK-1 levels were significantly higher in advanced-stage patients classified according to the BCLC (**F**) and UICC stages (**G**). (**H**,**I**) Serum DKK-1 levels were not related to hepatic reserve assessed by using the Child–Pugh score (**H**) and albumin-bilirubin (ALBI) grade (**I**).

**Figure 4 ijms-23-02801-f004:**
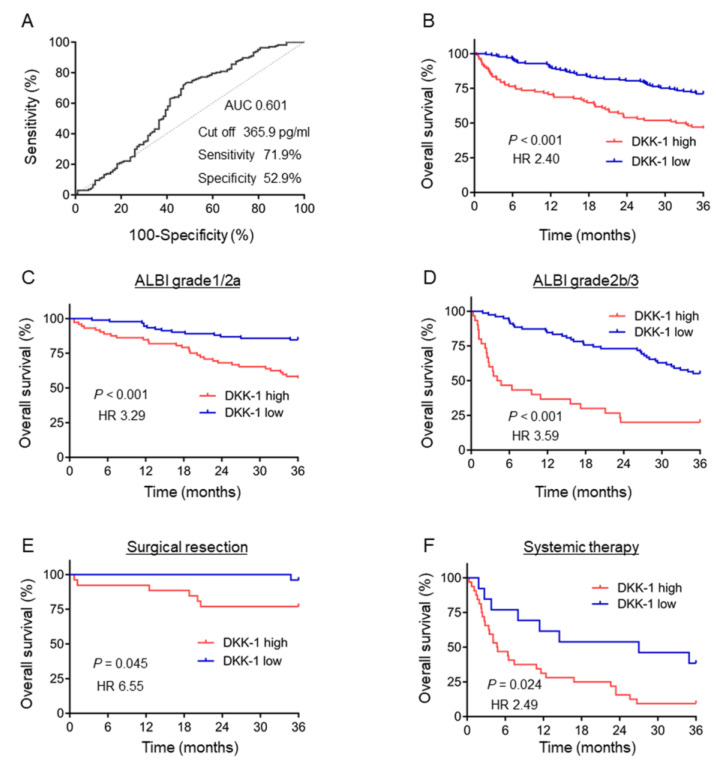
Overall (3-year) survival. (**A**) The receiver operating characteristic (ROC) curve analysis for survival showing the cutoff value of serum Dickkopf-1 (DKK-1) levels to be 365.9 pg/mL; area under the ROC curve (AUC) is 0.601, sensitivity is 71.9%, and specificity is 52.9%. (**B**) Kaplan–Meier survival analysis, indicating statistically significant correlation of elevated serum DKK-1 levels (≥365.9 pg/mL) with worse prognosis (*p* < 0.001). (**C**,**D**) In both albumin-bilirubin (ALBI) grade 1/2a (good liver function) and 2b/3 (poor liver function) cohorts, the high-serum DKK-1 group showed a statistically significant worse prognosis (*p* < 0.001). (**E**,**F**) The prognosis was significantly worse in patients with elevated DKK-1 levels in early stages treated by surgical resection (*p* = 0.045) as well as in those in an advanced stage treated by systemic therapy (*p* = 0.024).

**Figure 5 ijms-23-02801-f005:**
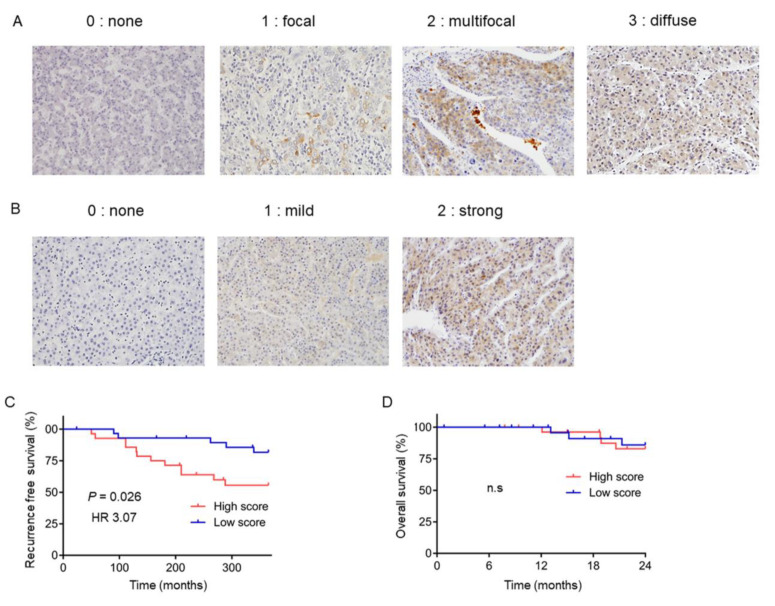
Dickkopf-1 (DKK-1) expression in resected hepatocellular carcinoma (HCC) tissues. (**A**,**B**) The staining area was evaluated and scored on four levels (none = 0, focal = 1, multifocal = 2, and diffuse = 3; **A**). The staining intensity was evaluated and scored on three levels (none = 0, mild = 1, and strong = 2; **B**), and the sum of both levels was used as the immunostaining score (0–5). (**C**) The Kaplan–Meier survival analysis indicates that DKK-1-high patients showed worse recurrence-free survival than DKK-1-low patients with statistical significance (*p* = 0.026). (**D**) The overall survival rate did not differ between the high- and low-DKK-1 groups.

**Figure 6 ijms-23-02801-f006:**
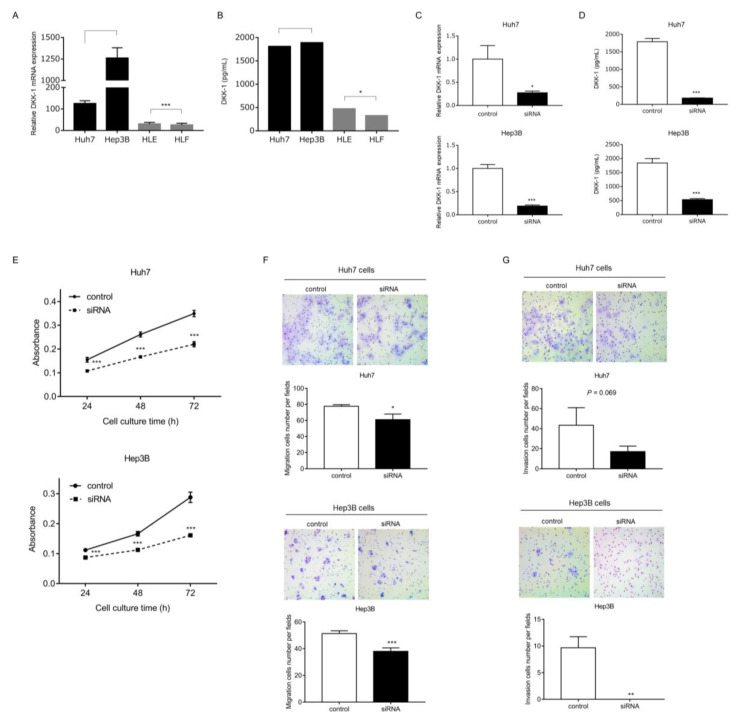
mRNA and protein expression of DKK-1 in HCC cell lines and in Huh7 and Hep3B cells knocked down for DKK-1 using siRNA. (**A**) DKK-1 mRNA expression was higher in Huh7 and Hep3B cells than in HLE and HLF cells (*p* < 0.001). (**B**) DKK-1 protein levels in the culture supernatant were significantly higher in Huh7 and Hep3B cells than in HLE and HLF cells (*p* < 0.05), as evaluated using ELISA. (**C**) *DKK1* mRNA levels were significantly decreased in Huh7 and Hep3B cells following siRNA transfection (* *p* = 0.013 and *** *p* < 0.001, respectively; *n* = 3 for each group). (**D**) DKK-1 protein levels in the culture supernatant were also significantly decreased in Huh7 and Hep3B cells following siRNA transfection (*p* < 0.001; *n* = 4 for each group). Proliferation, migration, and invasion in Huh7 and Hep3B cells. (**E**) The proliferation of Huh7 and Hep3B cells was significantly suppressed by *DKK-1* knockdown at 24, 48, and 72 h (*p* < 0.001 at 24, 48, and 72 h; *n* = 6 for each group. (**F**) Migration capacity was attenuated by DKK-1 knockdown in Huh7 and Hep3B cells with statistical or borderline significance (*p* = 0.017 and *p* < 0.001, respectively; *n* = 3 for each group). (**G**) Invasion capacity was attenuated by DKK-1 knockdown in Huh7 and Hep3B cells with borderline statistical significance (** *p* = 0.069 and *p* < 0.001, respectively; *n* = 3 for each group).

**Figure 7 ijms-23-02801-f007:**
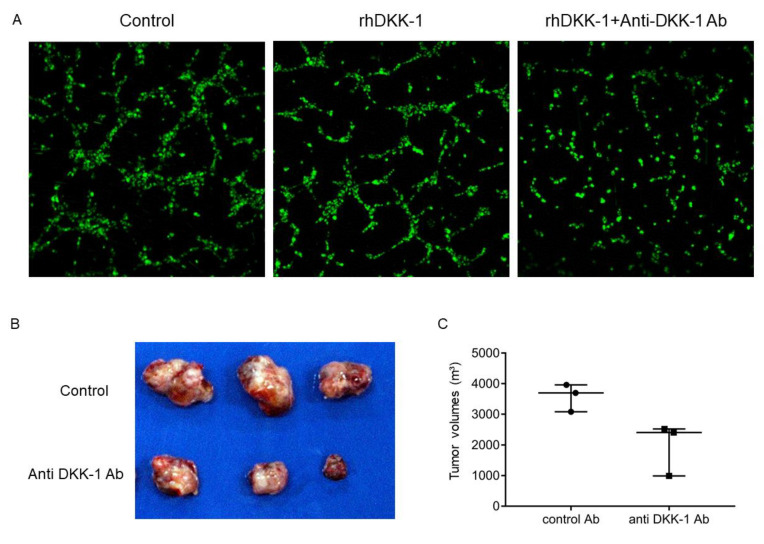
The effect of Dickkopf-1 (DKK-1) on angiogenesis and tumorigenesis. (**A**) Time-lapse image analysis of tube formation using human umbilical vein endothelial cells. DKK-1 treatment slightly accelerated tube formation compared with the control 2 h after treatment. This effect was abolished by the administration of anti-DKK-1 neutralizing antibodies. (**B**,**C**) Co-administration of anti-DKK-1 neutralizing antibody with Huh7 cells suppressed tumor growth in NOD/SCID mice compared with control IgG (*n* = 3 for each group).

**Table 1 ijms-23-02801-t001:** Characteristics of patients with hepatocellular carcinoma (HCC).

	Patients with HCC (*n* = 391)
Age, years	68 (61–74)
Sex, male, *n* (%)	260 (66.5)
Etiology, HCV/HBV/HBV+HCV/Alcohol/Others, *n* (%)	216 (55.5)/69 (17.6)/3 (0.8)/48 (12.3)/54 (13.8)
Child–Pugh, A/B+C, *n* (%)	286 (73.1)/105 (26.9)
ALBI grade, 1/2a/2b/3, *n* (%)	156 (39.9)/89 (22.8)/122 (31.2)/24 (6.1)
UICC (8th) stage, I/II/III/IV, *n* (%)	164 (41.9)/131 (33.5)/63 (16.1)/33 (8.4)
BCLC stage, 0/A/B/C/D, *n* (%)	54 (13.8)/149 (38.2)/109 (27.9)/59 (15.1)/19 (4.9)
Serum DKK-1 (pg/mL)	330.8 (272.8–409.5)
Serum AFP (ng/mL)	24.0 (10.0–200.0)
Serum PIVKA-II (mAU/mL)	50.0 (23.0–74.0)

Median (IQR).

**Table 2 ijms-23-02801-t002:** Characteristics of patients without hepatocellular carcinoma (HCC).

	Patients without HCC (*n* = 205)
Age, years	58 (50–64)
Sex, male, *n* (%)	114 (55.6)
Etiology, HCV/HBV/Alcohol/Others, *n* (%)	131 (63.9)/40 (19.5)/6 (2.9)/28 (13.7)
Liver cirrhosis, *n* (%)	28 (13.7)
Serum DKK-1 (pg/mL)	253.8 (204.3–331.5)

Median (IQR).

**Table 3 ijms-23-02801-t003:** Characteristics of patients in the groups with serum DKK-1 < 365.9 and ≥ 365.9 pg/mL.

	Serum DKK-1 < 365.9 (*n* = 169)	Serum DKK-1 ≥ 365.9 (*n* = 102)	*p*
Age, years	65.0 (61.0–74.0)	65.0 (58.0–74.0)	*n*.s.
Sex, male, *n* (%)	106 (62.7)	81 (79.4)	** 0.007
Etiology, HCV/HBV/HBV+HCV/Alcohol/Others, *n* (%)	111 (65.7)/24 (14.2)/3 (1.8)/20 (11.8)/11 (6.5)	37 (36.3)/28 (27.5)/0 (0.0)/16 (15.7)/21 (20.6)	*** < 0.001
UICC (8th) stage, I/II/III/IV, *n* (%)	75 (44.4)/64 (37.9)/24 (14.2)/6 (3.6)	28 (27.5)/27 (26.5)/25 (24.5)/22 (21.6)	*** < 0.001
BCLC stage, 0/A/B/C/D, *n* (%)	26 (15.4)/71 (42.0)/46 (24.2)/17 (10.1)/9 (5.3)	9 (8.8)/27 (26.5)/30 (29.4)/32 (31.4)/4 (3.9)	*** < 0.001
Serum AFP (ng/mL)	25.0 (11.0–170)	49.1 (10.0–1918)	*n*.s.
Serum PIVKA-II (mAU/mL)	40.0 (22.0–215)	397.0 (36.3–8555)	*** < 0.001

Median (IQR). ** *p* < 0.01; *** *p* < 0.001.

**Table 4 ijms-23-02801-t004:** Multivariate analysis of clinical characteristics for overall survival using the Cox proportional hazards model.

	Univariate	HR (95% CI)	Multivariate	HR (95% CI)
Sex	*n*.s.	—		
Age ≥ 65	*n*.s.	—		
Etiology	*n*.s.	—		
Child–Pugh (A/B+C)	*** < 0.001	2.08 (1.40–3.08)	*n*.s.	—
ALBI grade (1/2a/2b/3)	*** < 0.001	1.64 (1.34–2.00)	*** < 0.001	1.78 (1.45–2.17)
UICC (8th) stage (I/II/III/IV)	*** < 0.001	2.46 (2.02–3.00)	*** < 0.001	1.93 (1.55–2.40)
BCLC stage (0/A/B/C/D)	*** < 0.001	2.21 (1.76–2.49)	*n*.s.	—
DKK-1 (<365.9/≥365.9)	*** < 0.001	2.4 (1.64–3.54)	** 0.0015	2.02 (1.31–3.12)
AFP (<400/≥400)	** 0.0017	2.21 (1.46–3.35)	*n*.s.	—
PIVKA-II (<90/≥90)	*** < 0.001	3.76 (2.49–5.67)	*** < 0.001	2.59 (1.68–4.00)

** *p* < 0.01; *** *p* <0.001.

## Data Availability

Data are available from the corresponding author upon reasonable request.

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
