# Peer review of "Dickkopf-1 Promotes Angiogenesis and is a Biomarker for Hepatic Stem Cell-like Hepatocellular Carcinoma"

_ijms, 2022, doi:10.3390/ijms23052801_

Round 1

Reviewer 1 Report

Dear Editor, thank you so much for inviting me to revise this manuscript about HCC.

This study addresses a current topic.

The manuscript is quite well written and organized. English could be improved.

Figures and tables are comprehensive and clear.

The introduction explains in a clear and coherent manner the background of this study.

We suggest the following modifications:

  • Introduction section: although the authors correctly included important papers in this setting, we believe some studies should be cited within the introduction ( PMID: 34167433; PMID: 33907088 ; PMID: 33549983), only for a matter of consistency. We think it might be useful to introduce the topic of this interesting study.
  • Methods and Statistical Analysis: nothing to add.
  • Discussion section: Very interesting and timely discussion. Of note, the authors should expand the Discussion section, including a more personal perspective to reflect on. For example, they could answer the following questions – in order to facilitate the understanding of this complex topic to readers: what potential does this study hold? What are the knowledge gaps and how do researchers tackle them? How do you see this area unfolding in the next 5 years? We think it would be extremely interesting for the readers.

However, we think the authors should be acknowledged for their work. In fact, they correctly addressed an important topic in HCC, the methods sound good and their discussion is well balanced.

One additional little flaw: the authors could better explain the limitations of their work, in the last part of the Discussion.

We believe this article is suitable for publication in the journal although major revisions are needed. The main strengths of this paper are that it addresses an interesting and very timely question and provides a clear answer, with some limitations.

We suggest a linguistic revision and the addition of some references for a matter of consistency. Moreover, the authors should better clarify some points.

Author Response

Point-by-point responses to the reviewers’ comments

Reviewer #1

Dear Editor, thank you so much for inviting me to revise this manuscript about HCC. This study addresses a current topic. The manuscript is quite well written and organized. English could be improved. Figures and tables are comprehensive and clear. The introduction explains in a clear and coherent manner the background of this study. We suggest the following modifications:

We would like to thank the reviewer for the positive and constructive comments on our manuscript. As suggested by the reviewer, the manuscript has been comprehensively edited by a native English editor.

  1. Introduction section: although the authors correctly included important papers in this setting, we believe some studies should be cited within the introduction ( PMID: 34167433; PMID: 33907088 ; PMID: 33549983), only for a matter of consistency. We think it might be useful to introduce the topic of this interesting study.

Response

We would like to thank the reviewer for suggesting the citation of important papers. We have added the following sentences in the Introduction section wherein these references have been cited (page 1-2, lines 43–45):

“Thus, novel biomarkers are required to facilitate a better diagnosis of HCC [8]. Biomarkers for treatment are also currently being explored [9,10].”

  1. Discussion section: Very interesting and timely discussion. Of note, the authors should expand the Discussion section, including a more personal perspective to reflect on. For example, they could answer the following questions – in order to facilitate the understanding of this complex topic to readers: what potential does this study hold? What are the knowledge gaps and how do researchers tackle them? How do you see this area unfolding in the next 5 years? We think it would be extremely interesting for the readers.

Response

We would like to thank the reviewer for the valuable suggestion. We have revised the Discussion section and mentioned about the future perspectives in view of our findings. In particular, although alpha-fetoprotein is currently utilized as the first biomarker for identifying hepatocellular carcinoma patients who will receive survival benefits of ramucirumab therapy, the molecular mechanism underlying the correlation of the alpha-fetoprotein status with the anti-tumor effects of VEGFR2 blockade remains unclear. Because DKK-1 expression correlated positively with AFP in tumors and is involved in angiogenesis in the tumor microenvironment, serum AFP may be a surrogate marker of DKK-1, which facilitates angiogenesis in HCC. We have revised the Discussion section as follows to include the abovementioned points (page 14, lines 377–388):

“The results of this study suggest that serum DKK-1 affects angiogenesis in the microenvironment of HCC. In particular, although AFP is currently utilized as the first biomarker for identifying HCC patients most likely to receive the survival benefits of ramucirumab therapy [47], the molecular mechanism underlying the correlation of expression status of AFP with the anti-tumor effects of VEGFR2 blockade remains unclear. As DKK-1 expression correlated positively with that of AFP in tumors and is involved in angiogenesis in the tumor microenvironment, serum AFP may be a surrogate marker of DKK-1, which facilitates angiogenesis in HCC. In addition, another angiogenesis inhibitor, bevacizumab, which is used in combination with atezolizumab [48], and the multi-targeted tyrosine kinase inhibitors (TKIs), such as sorafenib, lenvatinib, regorafenib, and cabozentinib [49], also inhibit VEGF and VEGFRs. The relationship between these drugs and DKK-1 needs to be evaluated in the future.”

  1. One additional little flaw: the authors could better explain the limitations of their work, in the last part of the Discussion.

Response

We agree with the reviewer’s comment. We have mentioned the limitations of our study in the last part of the Discussion section (page 14-15, lines 389–395).

“This study has some limitations. The prognostic value of DKK-1 was evaluated in a retrospective manner, and the sample size of the cohorts was relatively small. The serum DKK-1 levels quantified by ELISA could be more specifically and sensitively measured if a chemiluminescent immunoassay system is appropriately developed. DKK-1 and cytoplasmic/nuclear β-catenin status could not be directly evaluated simultaneously due to the limited availability of small FFPE samples. Further studies are required to improve our knowledge regarding DKK-1 biology in HCC.”

Reviewer 2 Report

The manuscript by Suda et al titled “Dickkopf-1 Promotes Angiogenesis and is a Biomarker for Hepatic Stem Cell-like Hepatocellular Carcinoma” provides the functional role of DKK1 in HCC in vitro and in vivo.

 The manuscript was written well and easy to follow. Some additional data will enhance the overall impact of the findings.

Fig1.

  1. provide protein expression by western blot in EpCAM + and negative cells
  2. What are the other stem cell-specific antigens other than DKK and AFP expressed in HpSC-HCC compared to MH-HCC? Additional data on this would be good.

Fig.5 A-B:

If authors are able to provide DDK quantification by qPCR or western blot would be good

Fig 6:C: provide gene silencing efficiency in panel C (move that from supplemental to the main figure)

The migration and Invasion images should be in the main figure as well. One problem is I do not see a big difference in migration.  Same case with Invasion when I look at the images. So Panel C_F doesn’t really go well with the quantification data. Better to repeat this experiment to confirm this data.   

 Specifically, in Fig5F, quantification shows no invasive cells but images show no difference.

 In all figure legends, P-value is written but no information on  “n”

 Fig.7.

Provide some more data on tumor histology in addition to tumor size and volume.

Cell proliferation decreased? Ki 67/PCNA staining would be good.

What is the mechanism of DDK-induced cell/tumor growth?  Since it is a WNT signaling modulator, better to show nuclear beta-catenin levels in DKK silenced and control cells/ tumors and also levels of some of the WNT signaling targets.

Author Response

Reviewer #2

The manuscript by Suda et al titled “Dickkopf-1 Promotes Angiogenesis and is a Biomarker for Hepatic Stem Cell-like Hepatocellular Carcinoma” provides the functional role of DKK1 in HCC in vitro and in vivo.

We would like to thank the reviewer for the positive and constructive comments on our manuscript. Please find below our point-by-point responses to all the comments.

Major comments

  1. provide protein expression by western blot in EpCAM + and negative cells

Response

We would like to thank the reviewer for this suggestion. We have provided western blot data in Fig. 1 and added the relevant text in the manuscript.

  1. What are the other stem cell-specific antigens other than DKK and AFP expressed in HpSC-HCC compared to MH-HCC? Additional data on this would be good.

Response

We thank the reviewer for the pertinent query. We previously demonstrated that the expression of hepatic stem cell markers, EpCAM, AFP, DKK-1, DLK1, CD133, and CK19, is upregulated in HpSC-HCC compared with that in MH-HCC (Yamashita T et al. Gastroenterology 2009). We have added information on the stem cell marker status in the revised manuscript.

  1. 5 A-B: If authors are able to provide DDK quantification by qPCR or western blot would be good

Response

We agree with the reviewer’s comment. Unfortunately, because HCC samples were only available as FFPE tissues, we could not perform qPCR or western blot analysis owing to technical difficulties. We have described this limitation in the revised manuscript.

  1. Fig 6:C: provide gene silencing efficiency in panel C (move that from supplemental to the main figure)

Response

As you suggested, we have moved Supplementary Fig. 2 to Fig. 6 (top panel).

5Fig 6:The migration and Invasion images should be in the main figure as well. One problem is I do not see a big difference in migration.  Same case with Invasion when I look at the images. So Panel C_F doesn’t really go well with the quantification data. Better to repeat this experiment to confirm this data. Specifically, in Fig5F, quantification shows no invasive cells but images show no difference.

Response

In the hindsight, we agree that the reviewer has made a valid observation. We have, therefore, replaced the migration/invasion figures with more representative ones and have revised Fig. 6 accordingly. We have also cited previous reports about DKK-1 enhancing both the migration and invasion of HCC cell lines (Tung EK et al. Liver Int. 2011; Kuang HB et al. Front Biosci 2009; Chen L, et al. Mol Cancer. 2013) in the revised manuscript. We have moved Supplementary Fig. 2 to Fig. 6 (top panel).

6In all figure legends, P-value is written but no information on  “n”

Response

We would like to thank the reviewer for pointing this out. The information on “n” has been added to the main text and/or figure legends.

  1. 7.Provide some more data on tumor histology in addition to tumor size and volume.Cell proliferation decreased? Ki 67/PCNA staining would be good. What is the mechanism of DDK-induced cell/tumor growth?  Since it is a WNT signaling modulator, better to show nuclear beta-catenin levels in DKK silenced and control cells/ tumors and also levels of some of the WNT signaling targets.

Response

We respectfully agree with the reviewer’s comments. Regarding Ki-67 staining, we previously demonstrated the high-Ki-67 index in serum AFP-high HCC (Yamashita T, Hepatology 2014); we have cited this paper and have added the information in the revised manuscript (page 14, lines 342–345).

“We previously demonstrated that AFP elevation is accompanied with high KI-67 labeling index in surgically resected HCC specimens [50], which might be explained by the current findings of cell proliferation induced by DKK-1.”

However, unfortunately, we could not evaluate the beta-catenin status in this cohort because some samples were lost. We agree that information on beta-catenin staining and DKK-1 status would be very important to improve our understanding of the Wnt/beta-catenin signaling in HCC biology. We have mentioned these points as limitations in the Discussion section (page 14-15, lines 389–395).

“This study has some limitations. The prognostic value of DKK-1 was evaluated in a retrospective manner, and the sample size of the cohorts was relatively small. The serum DKK-1 levels quantified by ELISA could be more specifically and sensitively measured if a chemiluminescent immunoassay system is appropriately developed. DKK-1 and cytoplasmic/nuclear β-catenin status could not be directly evaluated simultaneously due to the limited availability of small FFPE samples. Further studies are required to improve our knowledge regarding DKK-1 biology in HCC.”

Round 2

Reviewer 1 Report

The authors modified the manuscript according to our suggestions.

We recommend Acceptance.

Author Response

Response to Reviewer 1

The authors modified the manuscript according to our suggestions.

We recommend Acceptance.

Response

Thank you for your careful review and suggestions. We appreciate your time and effort.

Reviewer 2 Report

 The authors have addressed my comments.

 There is a minor correction needed in the below paragraph.

There is no Figure 8. Figure 7 is described as Figure 8.

 Also, label what is the green color stands for in Figure7A.

----"Supplementation of recombinant DKK-1 slightly accelerated tube -----
---- with the control, whereas administration of anti-DKK-1 antibody inhibited tube 313
formation (Figure 8A;)..."

Author Response

Response to Reviewer 2

 The authors have addressed my comments.

 There is a minor correction needed in the below paragraph.

There is no Figure 8. Figure 7 is described as Figure 8.

 Also, label what is the green color stands for in Figure7A.

----"Supplementation of recombinant DKK-1 slightly accelerated tube -----
---- with the control, whereas administration of anti-DKK-1 antibody inhibited tube 313
formation (Figure 8A;)..."

Response

Thank you for noting our errors. In reviewing the legend, we found several additional small errors and corrected them. The revised text is highlighted in red font color.